# Specific Composition Diets and Improvement of Symptoms of Immune-Mediated Inflammatory Diseases in Adulthood—Could the Comparison Between Diets Be Improved?

**DOI:** 10.3390/nu17030493

**Published:** 2025-01-29

**Authors:** M. Dolores Guerrero Aznar, M. Dolores Villanueva Guerrero, Margarita Beltrán García, Blanca Hernández Cruz

**Affiliations:** 1Pharmacy Health Management Unit, Virgen Macarena University Hospital, 41009 Seville, Spain; lolavilguer@gmail.com (M.D.V.G.); margarita.beltran.sspa@juntadeandalucia.es (M.B.G.); 2Rheumatology Health Management Unit, Virgen Macarena University Hospital, 41009 Seville, Spain; blancae.hernandez.sspa@juntadeandalucia.es

**Keywords:** diet assessment, autoimmune disease, immune-mediated disease, immune-mediated inflammatory disease, diet, meta-analysis, multiple sclerosis, rheumatoid arthritis, biomarkers

## Abstract

Diet is considered a possible cofactor, which affects the immune system and potentially causes dysregulation of intestinal homeostasis and inflammation. This study aimed to review the quality of evidence on the effects of specific diet composition on symptoms of immune-mediated inflammatory diseases (IMIDs), including rheumatoid arthritis (RA), spondyloarthritis, multiple sclerosis (MS), inflammatory bowel disease (IBD) [remission maintenance of Crohn’s disease and ulcerative colitis], psoriasis and psoriatic arthritis in adult patients. We conducted a review of meta-analyses and Cochrane systematic reviews using PubMed and EMBASE, from inception to September 2024, and Google Scholar. The methodological quality of the meta-analyses was assessed using the AMSTAR 2 rating system. Three Cochrane systematic reviews and eight meta-analyses were evaluated. Some specific composition diets have been shown to reduce the symptoms of RA, IBD, and MS and improve activity parameters in IBD and RA, with critically low or low levels of evidence. The reduction in inflammatory biomarker levels is unclear. This review summarizes the global evidence for specific dietary interventions, mostly with anti-inflammatory properties due to their components, to improve IMID symptoms, clarifying the weaknesses of clinical trials and dietary meta-analyses with critically low or low levels of evidence; and shows the need to use indices such as the Dietary Inflammatory Index, which allows diets to be classified by their pro-inflammatory or anti-inflammatory food content, to better compare diet groups in clinical trials. The difficulty of obtaining high-level evidence from dietary studies is apparent and may delay the application of the results. Clinicians should be aware of the role of diets with anti-inflammatory properties as a complement to pharmacological treatments in IMIDs.

## 1. Introduction

Immune-mediated inflammatory diseases (IMIDs) involve a large, uncontrolled immune response characterized by acute or chronic inflammation that can affect any organ system. IMIDs share inflammatory pathways and their main characteristic is that an imbalance in inflammatory cytokines determines their onset and development; although the characteristics of these mediators vary among IMIDs. Furthermore, each of these diseases has a different epidemiology and pathophysiology [1,2]. This group includes, among other disorders, rheumatoid arthritis (RA), ankylosing spondylitis (AS), psoriasis, psoriatic arthritis (PsA), Crohn’s disease (CD), ulcerative colitis (UC) multiple sclerosis (MS), systemic lupus, type 1 diabetes, and asthma [2,3]. Over time, incidence and prevalence studies of IMIDs have been published, with considerable variation between different populations; in Western society, it ranges from 5% to 7% [2].

IMID patients have a higher risk for another IMID when compared with controls, indicating that the diseases may be related. In a nationwide study of the Danish population, 22.5% of patients with inflammatory bowel disease (IBD) also had at least one concurrent IMID [4].

Possible causes of IMIDs include genetic factors in an immune-susceptible host and environmental factors such as lifestyle, diet, drugs, infections, and agricultural and industrial practices [1,5]. Diet may contribute to the incidence of IMIDs through interaction mechanisms such as direct food antigens that cause intolerance/allergies, intestinal microbiome alterations, and intestinal permeability [5]. Abnormal inflammatory responses are closely associated with IMIDs, and chronic metabolic inflammation is associated with a Western diet, among other factors. A Western diet is associated with a risk of developing RA and intestinal bowel diseases [6,7].

### 1.1. Influence of Dietary Components on IMIDs. Table 1

#### 1.1.1. Diet Composition, Intestinal Microbiota, and Dysbiosis

The intestinal microbiota (IM) comprises all intestinal microorganisms, including genes, proteins, and metabolic products. IM imbalance, also known as dysbiosis, can influence the development of immune disorders through T lymphocyte (T cell) activity [20].

Microbial dysbiosis is the primary cause of local inflammation and IMIDs including colitis and IBD [21]. RA [9], MS [22,23], and other neurological disorders [18] can also be caused by gut dysbiosis. Changes in the gut microbiota are related to disease activity [9].

The composition of the IM can be affected by diet [24], and may influence susceptibility to intestinal inflammation and autoimmunity [25], as well as predict susceptibility to IMIDs such as RA [9]. In this context, diet can be a therapeutic strategy for preventing and treating RA, MS, and IBD by modulating IM [9,14,26].

Dietary Fibers are non-digestible carbohydrates that constitute an important energy source for the IM. Alterations in consumption modify the structure and function of IM within days [8]. The high Fiber food improves IM diversity as part of a balanced diet. Dietary Fiber consumption can improve remission rates in patients with CD and RA as part of a proper diet [9,10] and can help reduce joint pain in patients with RA [9].

The onset of IBD is associated with high carbohydrate intake, including sugar [14] and low vegetable intake [27], and reducing dietary carbohydrates can help improve the balance between IM and immune function in RA [9]. High glucose and fructose intake can reduce microbial diversity in the long term; therefore, food choices affect IM [18,28]. Excessive consumption of red meat can lead to changes in the microbiota, and increase the risk of RA [16], and along, with excessive dairy consumption may also increase the risk of acute flares in IBD and psoriasis [15].

*N*-glycolylneuraminic acid, which is present in red and processed meats, can be regulated by specific gut microbiota bacteria; however, *N*-glycolylneuraminic acid accumulation causes inflammation [29]. A high-sodium diet can also lead to dysbiosis, which promotes a micro-inflammatory state and autoimmune processes [9].

The microbiota modulates the gut immune system through omega-3 polyunsaturated fatty acids, which are useful for maintaining gut health. However, the composition of the microbiota is complex, and individualized diets must be designed to meet the needs of each patient [11].

Diet and IM interventions for the treatment of CD, RA, and MS are being studied [9,23,30]. Microbiome signatures are essential for controlling and preventing Autoimmune diseases (ADs) [22], and dietary patterns correlating with bacterial groups have been identified. Diet can affect the inflammatory responses in the gut through microbial mechanisms [14].

#### 1.1.2. Diet Composition and Intestinal Permeability

In addition to its essential role in regulating homeostasis, the intestinal barrier (IB) comprises the immune system of the intestinal mucosa [31]. Impaired IB function can activate autoimmune responses, thereby initiating and developing IMID [32]. As a consequence of a leaky gut, there is increased inflammation and risk of malabsorption of essential macro- and micronutrients. For example, vitamin D deficiency has been reported as a risk factor for some IMIDs such as RA and MS [5].

Omega-3 polyunsaturated fatty acids can help maintain IB integrity. A low omega-3/omega-6 fatty acid ratio promotes inflammation and increases the risk of RA [9]. Red and processed meats contain elevated levels of haem iron, which is correlated with alterations in the IB [15]. A high-fat diet can lead to dysregulated microbiota and metabolites, followed by IB dysfunction in UC [19]. Consumption of wheat and other cereals can contribute to chronic inflammation and ADs by increasing intestinal permeability [17]. Sourdough fermentation degrades wheat alpha-amylase/trypsin inhibitors and reduces pro-inflammatory activity [12].

IB can be negatively affected by alcohol and food additives, such as sweeteners, emulsifiers, and advanced glycation end products produced during food processing [18].

#### 1.1.3. Diet Composition, Inflammation, and Immune System

The immune system contains the highest number of energy-consuming cells in the body [5]. Nutrition influences the inflammatory cascade at the molecular level and IM [33]. Both pro- and anti-inflammatory signals, which can modulate gut immune responses, can be generated by dietary components that directly interact with the innate and adaptive immune systems [34].

A study in older adults above a healthy weight found that meals that differ in their nutritional composition and inflammatory potential do not differentially affect circulating cytokines after taking it only once, long-term dietary patterns are important [35]. Red meat consumption has been associated with an inflammatory pattern, characterized by an increase in IL-6 and IL-8. Frequent consumption of sweeps has been associated with IL-8 level increase, conversely, higher intake of peeled fruits was correlated with lower levels of IL-6. Additionally, IL-6 and IL-8 formed a cluster that also included IL-1b and TNF-α [13].

Potential interactions between diet and autoimmunity have been studied by creating a database of overlap between human autoimmune epitopes and food epitopes. The epitopes implicated in MS and RA have been identified in commonly consumed foods. Frequent consumption of foods containing epitopes implicated in these diseases may exacerbate the symptoms [36].

In addition to interacting at the molecular level, diet indirectly affects inflammation by modulating the IM. An increase in antigen-specific regulatory T cells may occur rapidly in response to alterations in the IM. Food contains components that can act as substrates for the microbiota and High-Fiber diets, have been shown to have a protective effect in animal models of arthritis. In addition, IM metabolites with immunoregulatory properties include short-chain fatty acids, protein catabolites, and vitamins [22,37].

### 1.2. Dietary Inflammatory Index (DII)

Changes in inflammatory biomarkers are important when comparing the effectiveness of diets with specific compositions for symptom improvement in patients with IMIDs. When comparing groups of diets with different compositions, it is important to know the potential pro- or anti-inflammatory capacities of all the components. The DII is a literature-derived population-based dietary score used to assess the inflammatory potential of an individual’s overall diet. The articles assessed the association of one or more of 45 food and nutrient parameters with six inflammatory biomarkers (IL-1beta, IL-4, IL-6, IL-10, tumor necrosis factor-alpha, and C-reactive protein -CRP- levels) (Figure 1).

The inflammatory potential for each food parameter was scored +1, −1, or 0 if it increased, decreased, or had no effect, respectively, on these inflammatory biomarkers. All of the “food parameter-specific DII scores” were added through a specific procedure to calculate the “overall DII score” (maximum pro-inflammatory diet DII score: +7.98, maximum anti-inflammatory DII score: −8.87, and median: +0.23) [38,39]. DII and energy-adjusted DII (E-DII) have been used in numerous studies in recent years [39].

### 1.3. Obesity

Obesity is an epidemic in the Western world and is defined as a body mass index (BMI) > 30 kg/m^2^. The main characteristics of obesity include accumulation, and rearrangement of different types of body fat, subclinical chronic inflammation, and metabolic dysfunction. Obesity is associated with RA, CD, MS, and psoriasis. Immunoregulatory effects are achieved through caloric restriction [7,23]; however, adherence to an appropriate diet is a major obstacle.

The odds of RA remission decrease with obesity. Overweight and obese individuals have a 1.27-fold increased risk for RA [40]. Obesity negatively affects disease activity in axial spondyloarthritis (Ax-SpA) and RA [41,42] and reduces the probability of an antitumor necrosis factor response in RA [43]. Weight loss affects disease activity in SpA [44], and non-pharmacological and non-surgical interventions for weight loss have been associated with reduced severity of psoriasis in overweight or obese patients [45].

The IM depends on dietary patterns and is individualized. Weight loss and decreased levels of inflammatory markers have been linked to different microorganisms, and the Prevotella-to-Bacteroides ratio has been used to predict successful body weight and fat loss [46].

We have summarized the results of the influence of specific dietary components on the reduction of IMID symptoms. However, the analysis of the practical application of these results using specific composition diets is still evolving and, due to the great heterogeneity of the studies, it has been difficult in the past to draw concrete conclusions, with sufficient evidence to be applied.

We hypothesized that a specific diet composition would influence the improvement of IMID symptoms. Consequently, this study aimed to review the quality of evidence and the results on the effects of specific diet composition on the symptoms of some IMIDs (RA, PsA, SpA, MS, IBD [remission maintenance of CD, UC], and psoriasis) in adults. Cochrane systematic reviews (Cochrane SR), and meta-analyses were evaluated, focusing on intervention studies and assessing pathological activity rates, patient-reported outcomes such as pain and fatigue scales, and inflammatory biomarker changes.

## 2. Methods

### 2.1. Search Strategy

We followed a standardized methodology according to the Preferred Reporting Items for Systematic Reviews and Meta-Analyses (PRISMA) 2020 [47,48]. We conducted a systematic search of meta-analyses and Cochrane systematic reviews using the electronic databases PubMed and EMBASE, from inception to September 2024, and Google Scholar. No date restrictions were used. See Appendix A for the complete search strategy per Data-Base. Only English-language articles were included, and duplicate articles were excluded.

### 2.2. Study Selection

Two authors used the eligibility criteria described below to screen studies for inclusion based on title and abstract, reviewers performed the screening independently. The studies that passed initial screening were studied based on the full text, to check inclusion and exclusion criteria.

### 2.3. Eligibility Criteria Used for Literature Search and Screening

Inclusion criteria: Eligible studies included adults with IMIDs (RA, PsA, SpA, MS, IBD [remission maintenance of CD, UC], and psoriasis). Intervention: Diet. Control: Placebo/other control interventions. Study types: Systematic reviews/meta-analyses focusing on intervention studies. Language: English.

Exclusion criteria: Studies including pediatric patients, irritable bowel syndrome, preoperative nutritional interventions, enteral or parenteral nutrition, fasting-mimicking diets, weight loss, exclusion diets + partial enteral nutrition, and studies that sought the effects of certain foods or nutrients (supplements, nutraceuticals, micronutrients, and vitamin D) without a clear diet designation.

Limitations: Not all IMIDs (systemic lupus, type 1 diabetes, asthma) [3] were included or comprehensively reviewed.

### 2.4. Data Extraction

Two researchers independently assessed all articles. The articles were reviewed, and the data were tabulated for the selected variables. A third author helped reach the consensus in cases of disagreement.

### 2.5. Outcomes Measures

MS: Change in Expanded Disability Status Scale (EDSS) [49,50], (ranges from 0 [no neurological abnormality] to 10 [death due to MS]), Multiple Sclerosis Quality of Life-54 [51], and the Modified Fatigue Impact Scale (MFIS) [52].

IBD: Change in clinical disease activity index (CDAI) [53], functional gastrointestinal symptoms (FGSs), and quality of Life (QoL).

RA: Change in disease activity score 28 (DAS 28) [54], pain scales, and all relevant efficacy outcomes.

Psoriatic arthritis, SpA, and psoriasis: all relevant efficacy outcomes.

Inflammatory biomarkers: Changes in CRP and interleukin [IL], fecal calprotectin [FC], and albumin. 

Standardized mean difference (SMD), Odds Ratio (OR), or Relative Risk (RR), were analyzed [55].

Safety: Number of severe adverse events associated with dietary intervention during the follow-up period.

### 2.6. Quality of Meta-Analyses

We assessed the methodological quality of each meta-analysis (MA) using AMSTAR 2, a validated tool for evaluating systematic reviews and meta-analyses [56]. It consists of 16 items that assess methodological quality as one of four grades (high, moderate, low, or critically low), and includes ratings for the quality of the search, reporting, risk of bias, and transparency of the MA.

## 3. Results

### 3.1. Characteristics of Eligible Studies

A flowchart is shown in Figure 2 with details of the included studies and selection processes.

### 3.2. Cochrane SRs

Appendix A includes a description of clinical trials on diets evaluated in the Cochrane SRs.

#### 3.2.1. Cochrane SR and Meta-Analyses on the Outcomes of Diet on Rheumatic and Musculoskeletal Diseases

In a systematic review and meta-analysis (SR-MA) of the effects of diet on rheumatic and musculoskeletal diseases (Gwinnut 2022), several rheumatic pathologies were evaluated. In this SR-MA [57] 1 MA (Cramp 2013), and, 12 randomized controlled trials (RCTs), (Appendix A), 5 no randomized trials, 1 single-arm study, and 1 extension to RCT on experimental diets on RA were studied, reporting no effect on the majority of outcomes assessed. The evidence for most dietary exposure in RA was classified as low or very low because of the small number of studies and sample sizes. 1 MA, 1 RCT, and non-randomized trials studied the Mediterranean diet for RA. The Cramp MA, 2013 (Cochrane Database Syst Rev) -AMSTAR 2 Hight Quality- [58], reported no significant effect of the Mediterranean diet on fatigue based only on the analysis of Skoldstam et al., 2003 RCT -SMD 0.37, 95% CI −0.18 to 0.93-. This RCT reported a large effect of the diet on pain and a small effect on disease activity. Dietary biomarkers were evaluated in the trial, with significant differences in CRP levels between the groups, but not in erythrocyte sedimentation rate (ESR) levels.

#### 3.2.2. Cochrane SR on the Outcomes of Diets in IBD

A 2019 Cochrane SR by Limketkai et al. [59], evaluated RCTs that compared the intervention diet versus the control diet for the induction, maintenance, and clinical relapse of remission in active CD and UC, and in inactive CD and UC (Appendix A). Studies that exclusively focused on enteral nutrition, oral nutrient supplementation, medical foods, probiotics, or parenteral nutrition were excluded. Induction of remission refers to a therapeutic reduction in bowel symptoms below the established scores. Maintenance of remission refers to a decrease in symptoms over time owing to diet. Clinical relapse is defined based on the symptom scores.

The 18 studies included 1878 randomized participants. The evaluated dietary interventions consisted of dietary restriction or the exclusion of dietary components thought to induce IBD symptoms or inflammation (Appendix A). This wide variety of diets generally did not allow data to be studied in a grouped manner.

Active CD. Insufficient evidence was found regarding the exclusion diets and improvement of clinical remission rates in active CD, and the evidence was assessed as very low. In terms of quality of life (QoL) outcomes, the mean difference in the Inflammatory Bowel Disease Questionnaire (IBDQ) score between the symptom-guided and control diets (Daniel study, 2007) showed very low certainty. After the analysis of surrogate biomarkers of inflammation, it was uncertain whether highly restricted organic diets led to a difference in CRP or ESR, based on the analysis of the study by Bartel (2008) (Appendix A).

Inactive CD. There was little or no difference in the clinical relapse rates between the exclusion diets (Appendix A) and the control diets in inactive CD. The evidence was assessed as low or very low comparing low disaccharide, low grain, low saturated fat, low red, and processed meat diets with a control diet. For inflammatory biomarkers, Riordan (1993) assessed the effect of symptom-guided diets on CRP levels after 24 months. There were no significant differences between the groups. Evidence from the analysis of Jones 1985, and Jordan 1993 studies (Appendix A) shows that the effect of symptom-guided diets on the ESR is uncertain.

Active UC. There is a very low level of evidence that symptom-based diets improve clinical remission rates in patients with active UC. The level of evidence for the Alberta-based anti-inflammatory diet or the carrageenan-free diet (Appendix A), reducing clinical relapse rates in inactive UC, was also very low.

Inactive UC. They did not find sufficient evidence to demonstrate that a milk-free diet reduces clinical relapse rates in patients with inactive UC. No significant changes were found in inflammatory biomarkers or the short IBDQ (SIBDQ) with the carrageenan-free diet (Bhattacharyya 2017) (Appendix A). None of the studies on diets for UC patients reported any adverse events.

The number of studies and patients was too low to allow a comparison of diet variables with similar characteristics. Limketkai’s 2019 SR findings showed that the effects of dietary interventions on CD and UC were uncertain. The evidence was very low due to the scarcity of data coming from heterogeneous studies that were difficult to group. Therefore, more RCTs and a consensus on the composition of these interventions are needed.

Limketkai et al., in a 2022 SR-MA with a similar publication title to 2019, Limketkai et al. Cochrane Review studied solid food diets for IBD remission induction or maintenance. They concluded that partial enteral nutrition was similar to exclusive enteral nutrition and potentially benefits CD remission induction and maintenance [60]. Total and partial enteral nutrition were exclusion criteria in the review carried out in the article being developed.

#### 3.2.3. Cochrane SR on the Outcomes of Diets and MS

A 2019 Cochrane SR on diet and MS [61] evaluated three RCTs focused on dietary interventions in MS-related symptoms: the Paleolithic diet (Irish 2017), a low-fat plant-based diet (Yadav 2016), and the co-supplemented hemp seed and evening primrose oils and advised hot nature diet (Rezapour-Fioruzy 2013) (Appendix A).

The study by Irish et al. (2017), was RCT involving patients with RRMS that compared the modified Palaeolithic diet (complete abstinence from products containing gluten, dairy, potatoes, and legumes) to the usual diet. Within 3 months, the fatigue severity scale (FSS) was significantly reduced, and physical and mental quality of life (MSQoL-54) improved in the intervention group compared with those in the control group. No adverse events were reported.

Yadav et al. conducted a randomized single-blind trial with the relapsing-remitting multiple sclerosis (RRMS) very low-fat plant-based diet (restrictions: meat, fish, eggs, dairy products, and vegetable oils such as corn and olive oil were prohibited) and control group (typical physician recommendation for MS). Fatigue was statistically significantly improved in the diet group compared with that in the control group, according to the FSS and MFIS at 12 months.

The Rezapour Fioruzy study was a parallel-group, randomized double-blind trial with RRMS participants (co-supplemented hemp seed and evening primrose oils with or without advised hot nature diet, versus supplemented olive oil -placebo group). The hot nature diet restrictions were low intake of cholesterol, hydrogenated or trans fatty acids, and saturated fats; reduction of sugar, refined starch, and dairy products with honey or dates; and elimination of foods with a cold nature. After 6 months, significant improvements in EDSS and the relapse rate were found in the intervention group, while the control group showed a border significant decrease in relapse rate. None of the three studies in the CR group showed significant adverse effects.

The Cochrane review reported a high risk of bias for these three trials (missing outcome data, lack of or inadequate blinding, and random dietary intervention) and concluded that there was insufficient evidence to determine whether dietary interventions impacted MS-related outcomes.

### 3.3. Meta-Analyses

Information regarding the included meta-analyses is presented in Table 2, the AMSTAR 2 ratings in Table 3, and information regarding the trials included in these meta-analyses in Appendix A.

The AMSTAR 2 ratings are shown in Table 3: critically low (three studies, 37.5%) and low (five studies, 62.5%). The main limitations are the absence of blinding in the therapeutic maneuver, outcome measures, and therapeutic compliance measures, as well as the great variability in the evaluated diets and inflammatory effects.

Meta-analyses on diet and IMID symptoms are summarized below.

#### 3.3.1. RA, PsA, and SpA

Schönenberger, et al. in a 2021 SR-MA [62] (Mediterranean, vegetarian, vegan, and ketogenic diets–all considered anti-inflammatory diets/control), evaluated seven RCTs (Appendix A) and concluded that anti-inflammatory diets resulted in significantly lower pain than ordinary diets in patients with RA. Subgroup analysis showed that the Mediterranean diet tended to improve pain better than vegetarian or vegan diets. All the studies had a high risk of bias. Owing to the lack of blinding, the pain data (patient-reported outcomes) could have been biased, and the evidence was very low. Five of the seven studies were evaluated to summarise the effects of anti-inflammatory diets on CRP levels. There were no significant differences between the groups. Patients who followed an anti-inflammatory diet lost more weight than those in the control group, and their BMI decreased. RCTs with a longer intervention period (13 months) tended to have significant effects, and those with a higher baseline BMI appeared to have a greater improvement in pain. These data are shown in Table 2.

Genel et al. in SR-MA from 2020 [63] evaluated the effects of a low-inflammatory diet intervention in adult patients with RA, osteoarthritis (OA), or seronegative arthropathy (psoriatic, reactive, AS, or IBD-related). Only one RCT and one prospective trial of patients with RA were evaluated (Appendix A), with a very low level of evidence; and no validated tools were used to assess dietary compliance in these trials. Genel et al. highlighted the differences between OA and RA in terms of the efficacy of dietary interventions; patients with RA who followed an anti-inflammatory diet showed greater improvements in pain and physical function, and no statistically significant benefits were observed in patients with OA. RCTs with significant effects were those with the longest intervention periods. Outcomes were recorded from 2 to 4 months and beyond, and patients in the intervention group with the highest baseline BMI showed a greater improvement in pain. A low-inflammatory diet can cause weight loss, and biomarker improvements can occur independently of weight changes. In Genel’s MA of inflammatory biomarker changes scores, Skoldstam’s CRP results on RA, and Schell’s and Dier’s IL-6 results on OA were utilized (Table 2).

Turk et al. in the SR-MA from 2023 [64] (11 RCTs) studied dietary interventions (gluten-free vegan, anti-inflammatory, and Mediterranean diets/controls) in patients with RA. Meta-analyses of the DAS28 and pain (4 RCTs, Appendix A) showed a very low level of evidence and significant symptom reduction when the diet was modified (Table 2). The anti-inflammatory diet achieved good results (reduced swollen joint count, tender joint count, pain, and DAS28) (Table 2). Changes in inflammatory biomarker scores were not evaluated in this MA.

**Table 2 nutrients-17-00493-t002:** Characteristics and some results of the meta-analyses included in the study.

Reference	Study Design	Studies	Pathology	Intervention/Exposure/Control	Outcomes	Main Results
(Genel et al., 2020) [63]	SR and MA	5 RCT (2 RA, 3 OA), 2 PPT (RA). 468 P. I: 259; C:226.	RA, OA, Seronegative arthropathies (psoriatic, reactive, ankylosing spondylitis or IBD-related)	Low-inflammatory diet, Anti-inflammatory diet, Mediterranean diet -caloric or non-caloric restricted-, to the usual diet.	Meta-analysis (*not including ketogenic diet*) of (1) weight change (2) CRP, Interleukin 6. (3) Pain score	(1) Weight SMD −0.45 (CI −0.71, −0.18). (2) CRP: −50.65 mg/L (−83.4, −18.26), RA patients. Inflammatory biomarkers (SMD −2.33. (CI −3.82, −0.84), *p* = 0.002, OA + RA patients. RA group (SMD −1.10 [95 % CI −1.71, −0.49], *p* = 0.0004 (3) Joint pain (SMD −0.98; CI −2.90, 0.93), *p* = 0.31. In a subgroup analysis pooled pain scores in a RA group were (SMD −2.81 CI −3.6, −2.02), *p* < 0.00001.
(Schönenberger et al., 2021) [62]	SR and MA	7 RCT. 326 P. (92% female P.).	RA	Mediterranean, vegetarian, vegan, and ketogenic diets, to usual diet.	Meta-analysis of RA pain, and CRP.	Anti-inflammatory diets resulted in significantly lower pain than ordinary diets (−9.22 mm; 95% CI −14.15 to −4.29; *p* = 0.0002. There were no significant differences in CRP levels (SMD −2.51 CI −6.10, −1.08).
(Turk et al., 2023) [64]	SR and MA	11 RCT.	RA	Dietary changes.	Meta-analyses of RA DAS28, and pain.	DAS28 was significantly improved in the group treated with diet (SMD −0.46 CI −0.91, −0.02), *p* = 0.04, with a very low level of evidence. Pain (SMD −1.23 CI −1.96, −0.5), *p* = 0.001.
(Comeche et al., 2020) [65]	SR and MA	31 research studies. MA: 10 trials: RCT, UNRCT, NRCT. 279 P.	IBD (CD, UC)	Low microparticle diet, semi-vegetarian diet (lacto-ovo-vegetarian diet- fish once a week and meat once every 2 weeks, both of them at half the average amount, bread rarely), IgG4-guided exclusion diet (CD), and others. to the usual diet.	Meta-analyses of CDAI for CD patients. FC, CRP, and ALB, for IBD patients.	A tendency to CDAI reduction (SMD −107.62 CI −182.12; −33.12), *p* < 0.01. No differences were observed for CRP (SMD −0.45 CI −1.06, 0.16), ALB (SMD 0.16 CI 0.09, 0.41), and FC (SMD −7.40 CI −54.12, 39.32).
(Zhan et al., 2018) [66]	SR and MA	2 RCTs and 4 before-after studies. 319 P.	IBD	LFD to the usual diet.	The effect of diets on diarrhea, abdominal bloating, abdominal pain, fatigue, and nausea.	Improvement in other symptoms: diarrhea response (OR: 0.24, 95% CI: 0.11–0.52, *p* = 0.0003), abdominal bloating (OR: 0.10, 95% CI: 0.06–0.16, *p* < 0.00001), abdominal pain (OR: 0.24, 95% CI: 0.16–0.35, *p* < 0.00001), fatigue (OR: 0.40, 95% CI: 0.24–0.66, *p* = 0.0003) and nausea (OR: 0.51, 95% CI: 0.31–0.85, *p* = 0.009). Very low level of evidence.
(Peng et al., 2022) [67]	SR and MA	4 RCTs and 5 before–after studies. 446 P. (96% in remission).	IBD	LFD to the usual diet.	FGSs. QoL -SIBDQ-(only 2 studies).	Overall FGSs (RR: 0.47, 95% CI: 0.33–0.66, *p* = 0.0000). QoL -SIBDQ (MD = 11.24, 95% CI 6.61 to 15.87, *p* = 0.0000). CD patients (RR: 0.44 CI 0.34–0.55) UC (RR: 0.43 CI 0.33–0.56). No statistically significant differences in normal stool consistency and mucosal inflammation.
(Guerrero Aznar et al., 2022) [68]	RR and MA	8 RCT. 515 P.	MS	Modified anti-inflammatory diet, Modified Mediterranean diet, Fasting-Mimicking-diet + Mediterranean diet, Modified Paleolithic diet -gluten-free-, Co- hot nature diet, very low-fat plant-based Diet, Supplemented Mediterranean-type diet, to usual diet.	The effect of some anti-inflammatory diets on EDSS, MFIS, and QOL.	MA: A trend of reduction in fatigue (MFIS) (308 pat.) SMD −2.033, 95%-CI (−3.195, −0.152), *p*-value: 0.0341; increase in QOL Physical (77 pat.), SMD 1.297, (0.2454, 2.3485). *p*-value of 0.01; and in QOL mental (44 pat.), SMD 1.1086, 95%-CI (0.6143, 1.6029). *p* < 0.0001. No significant effect on EDSS (337 pat.), and no severe adverse events.
(Snetselaar et al., 2023) [69]	SR and NWMA	12 RT. 608 P.	MS	Low-fat, Mediterranean, ketogenic, anti-inflammatory, Paleolithic, fasting, calorie restriction, to the usual diet.	The effect of diets on MFIS, and QOL.	NWMA: The Paleolithic (SMD −1.27; 95% CI −1.81 to −0.74), low-fat (SMD −0.90; 95% CI −1.39 to −0.42), and Mediterranean (SMD −0.89; 95% CI −1.15 to −0.64) diets showed greater reductions in fatigue compared with control. The Paleolithic and Mediterranean diets showed greater improvements in physical and mental QoL compared with the control. Very low level of evidence.

I: Intervention. C: Control. SR: Systematic review. MA: Meta-analysis. NWMA: Network Meta-analysis. P: Patients. SMD: Difference between means. OR: Odds Ratio. RR: Relative Risk. MD: Means Difference. PPT: Pre-Post trial. RCT: Randomized Controlled Clinical Trial. NRCT: Non-Randomized Controlled Clinical Trial. UNRCT: Uncontrolled and Non-Randomized Clinical Trial. PPT: Pre–post-trial. CDAI: Crohn’s Disease Activity Index. FC: Fecal Calprotectin. CRP: C-Reactive Protein. ALB: Albumin. MFIS: Modified Fatigue Impact Scale. QOL: Quality of life. EDSS: Expanded Disability Status Scale. GSRS: Gastrointestinal symptom rating scale. IBS QoL: Irritable bowel syndrome quality of life DAS28: Disease activity index on a 28 joint count. SJC: Swollen joint count. TJC: Tender joint count. HAQ: Health assessment questionnaire. FGS: functional gastrointestinal symptoms. SIBDQ: Short IBD questionnaire.

**Table 3 nutrients-17-00493-t003:** Quality assessment of included meta-analyses according to AMSTAR 2 rating system.

References	ITEMS																
	1	2	3	4	5	6	7	8	9	10	11	12	13	14	15	16	Final rating
(Genel et al., 2020) [63]	Y	Y	Y	pY	Y	Y	Y	Y	Y	N	N	Y	N	N	NA	Y	Y:10, pY:1, N:4, NA:1
(Schönenberger et al., 2021) [62]	Y	Y	Y	pY	Y	Y	Y	Y	Y	N	Y	Y	Y	N	NA	Y	Y:12, pY:1, N:2, NA:1
(Turk et al., 2023) [64]	Y	pY	Y	pY	Y	N	pY	y	Y	N	Y	Y	Y	N	Y	Y	Y:9, pY:4, N:3
(Comeche et al., 2020) [65]	Y	pY	Y	pY	Y	Y	Y	Y	Y	Y	Y	Y	Y	N	NA	Y	Y:11, pY:3, N:1, NA:1
(Zhan et al., 2018) [66]	Y	pY	Y	pY	N	Y	pY	Y	pY	N	Y	Y	N	N	NA	Y	Y:8, pY:4, N:3, NA:1
(Peng et al., 2022) [67]	Y	Y	Y	pY	Y	Y	pY	Y	Y	N	Y	Y	Y	Y	y	Y	Y:13, pY:2, N:1
(Guerrero Aznar et al., 2022) [68]	Y	pY	Y	pY	Y	Y	Y	Y	pY	N	Y	Y	Y	Y	NA	Y	Y:10, pY:3, N:2, NA:1
(Snetselaar et al., 2023) [69]	Y	Y	Y	Y	Y	Y	Y	Y	Y	N	N	Y	N	N	Y	Y	Y:12, N:4

AMSTAR 2: 1, PICO description (Population. Intervention. Comparison or Control. Outcome -desired or of interest-); 2, protocol registered before the commencement of the review; 3, study design included in the review; 4, adequacy of the literature search; 5, two authors study selection; 6, two authors study extraction; 7, justification for excluding individual studies; 8, included studies descripted in detail; 9, risk of bias for the single studies being included in the review; 10, source of funding of primary studies; 11, appropriateness of meta-analytical methods; 12, the impact of risk of bias of single studies on the results of the meta-analysis; 13, consideration of risk of bias when interpreting the results of the review; 14 explanation and discussion of the heterogeneity observed; 15, assessment of presence and likely impact of publication bias; 16, funding sources and conflict of interest declared. Abbreviations: Y, yes; pY, partial yes; N, no; NA, not applicable. High quality: 0–1 non-critical weakness. Moderate: >1 non-critical weakness. Low: 1 critical flaw with or without non-critical weaknesses. Critically low: >1 critical flaw with or without non-critical weaknesses. Critical items (2, 4, 7, 9, 11, 13, 15). bias.

No specific systematic reviews or meta-analyses were found on diet and PsA or AS. A systematic review (2022) updated the evidence on the efficacy and safety of non-pharmacological treatments to inform the Assessment of Spondyloarthritis International Society-European Alliance of Associations for Rheumatology recommendations for SpA management. No conclusions were reached because the studies were heterogeneous in terms of the type, duration, and frequency of treatment [70].

#### 3.3.2. IBD

In a 2020 SR-MA on IBD by Comeche [65] (Table 2), 31 research studies were selected for qualitative synthesis, of which only 10 were selected for MA. They evaluated different predefined diets for patients with active IBD and those in remission, based on the decreased consumption of some types of pro-inflammatory foods or an increase in others to improve IM, which was compatible with other medical treatments. Crohn’s Disease activity index significantly improved in the group treated with diet but with reasonable doubt due to the high heterogeneity of the data. No differences were observed in the levels of some inflammatory markers (albumin, CRP, and FC). Improvement was observed in the low microparticle, semi-vegetarian, and immunoglobulin-exclusion diets (IGED). A low fermentable oligo-, di-, monosaccharides, and polyols (FODMAP) diet decreased symptoms in most individuals with IBD, however, they can also produce deficiencies when maintained for long periods, which can be avoided by supplementation. Specific carbohydrates and low-FODMAP diets can contribute to vitamin D deficiency. IGEDs improve IBD activity, and a gluten-free diet, despite low adherence, can be useful for managing IBD.

In the 2018 Low FODMAP diet MA on IBD by Zhan, two RCTs and four before and after studies [66] (Table 2), showed significant improvement in diarrhea, satisfaction with gut symptoms, abdominal bloating, abdominal pain, fatigue, and nausea, except for the constipation response, with very low level of evidence.

In the Low FODMAP diet MA on IBD by Peng in 2022 [67] (Table 2), four RCTs and five before and after studies showed a benefit in functional gastrointestinal symptoms (FGSs) and QoL but no improvement in stool consistency or mucosal inflammation in patients with IBD. Symptom improvement was observed in bloating, wind or flatulence, borborygmi, abdominal pain, and fatigue or lethargy, but not in nausea and vomiting. Only two studies assessed QoL using the SIBDQ, showing a reduction in the total SIBDQ score. However, two studies evaluated QoL and did not observe significant differences using the Irritable Bowel Syndrome QoL.

Changes in the inflammatory biomarker scores were not evaluated by Zhan et al. and Peng et al. in their meta-analyses.

#### 3.3.3. MS

In a 2022 MA on the effect of diet on MS, Guerrero et al. [68], evaluated eight RCTs with different predefined diets (Table 2). Dietary interventions were associated with reduced fatigue, increased QoL, no significant effect on the EDSS scores, and no severe adverse events. (Table 2). The rationale for dietary intervention was to control inflammation and oxidative stress. Studies with the best results excluded or reduced saturated fats, white flour, dairy, sugar, and meat and recommended the intake of sufficient fruits, vegetables, and fish. A 12-h fast at night provided good results. One study examined in this MA (Bohlouli, 2021) [71] showed a relationship between the DII and MFIS. The DII predicted MFIS in the modified Mediterranean diet group. The control diet (traditional Iranian diet) was adjusted for energy intake to avoid unexpected changes in body weight. Only two RCTs (Mousavi et al. and Rezapour et al.), found increased interleukin 4 (an anti-inflammatory cytokine) in the intervention group. Among the eight diets investigated, the modified Mediterranean diet was easier to maintain (Mousavi et al. study on diet adherence: 96.15%, Katz Sand: 90.3%), compared to restrictive diets (Irish et al.: 50%, Bohlouli et al.: 75.5%).

The SR and network MA by Snetselaar et al., in 2023 [69] jointly studied 10 RCTs and one uncontrolled parallel-group clinical trial (modified Palaeolithic diet/low saturated fat diet) and established diet groups that were considered similar, such as the Mediterranean diet (Mediterranean and modified Mediterranean diets). It was concluded that dietary interventions such as Palaeolithic and Mediterranean diets can reduce MS-related fatigue and improve physical and mental QoL (Table 2). Changes in the inflammatory biomarker scores were not evaluated.

Both studies [68,69] show that it is difficult to reach a sufficient level of evidence in dietary studies to make recommendations. The potential of dietary intervention and the benefit-risk ratio must be considered [68].

#### 3.3.4. Psoriasis

No MA or Cochrane SR were specifically identified for diet and psoriasis. A 2018 systematic review of the dietary recommendations for adults with psoriasis or PsA by the medical board of the National Psoriasis Foundation [72] updated the literature based on prior systematic reviews. They concluded that dietary interventions could be added to standard medical therapies to reduce disease severity. Dietary weight reduction using a hypocaloric diet is recommended for overweight or obese patients with psoriasis.

## 4. Discussion

This review of Cochrane SRs and meta-analyses, mostly RCTs, provides a critical assessment of the literature on the effects of diet on disease symptoms in patients with IMID. Most intervention diets studied restricted foods with inflammatory effects. Reduced pain, fatigue, and quality of life were observed in the patients with IMID and specific dietary interventions, with critically low or low levels of evidence.

The anti-inflammatory effects of specific nutrients are mediated through their ability to modulate the immune system [5]. The consumption of foods with high cholesterol, sugars, saturated fats [73], wheat, and other cereals [17] is associated with inflammation. This does not apply to the consumption of foods containing Fiber; vitamins; and low levels of alcohol, herbs, and spices. The magnitudes of these associations are small [73].

### 4.1. Efficacy of Diets with Specific Compositions to Reduce IMID Symptoms. Quality of Evidence

Among the limitations of this global review, we found that the meta-analyses reviewed included non-blinded studies, with small samples and predominantly of poor quality. The methodological quality of the meta-analyses included in the review was assessed using the AMSTAR 2 rating (Table 3). The selected clinical trials were not homogeneous (dietary interventions/complete diets, unknown and possibly diverse DII of diets, patients with different baseline characteristics, and various types of habitual diets by country). Similarly, the equation of Harris-Benedict was used to determine energy needs in some trials; however, intolerance tests were not performed to determine the baseline characteristics of the patients. In some studies, the level of adherence to a specified diet was extremely low, and patient preferences were not studied. Dietary adherence is not usually quantified as a primary MA result, with some exceptions [68,69]. Different methods have been used to monitor compliance, all of which are based on self-reported mean adherence. Among the various diets investigated, the modified Mediterranean diet was easier to maintain (90.3–96.15%) than restrictive diets (50–75.5%).

Safety, defined as the number of severe adverse events associated with dietary interventions within the follow-up period in clinical trials, usually works in favor [59,68,74].

This review of the influence of diets with a specific composition on the improvement of symptoms of IMID shows that dietary interventions have considerable potential. As the available evidence indicates, diets that reduce inflammatory foods should play a role in the treatment of IMIDs. Some specific composition diets have been shown to reduce symptoms in RA, IBD, and MS; improve some activity parameters in IBD and RA; and improve QoL in MS and IBD, with critically low or low levels of evidence. Based on a recent systematic review of RCTs, there is evidence that some interventions, including diet or dietary supplements, might also have positive effects on DAS28, reducing the disease activity score in RA. Positive findings need to be confirmed in future high-quality studies [75].

The European Alliance of Associations for Rheumatology recently published recommendations regarding lifestyle and diet for patients with rheumatic and musculoskeletal diseases, who must be informed about no specific beneficial food types and should aim for a healthy weight [57].

The British Dietetic Association consensus guidelines state that there is insufficient evidence to recommend an anti-inflammatory diet for maintaining IBD remission. However, they did not consider the anti-inflammatory potential of most of the diets evaluated in the consensus, which were not referenced as anti-inflammatory diets [76].

Similarly, a systematic review of the Crohn’s disease Exclusion Diet (CDED), which reduces exposure to individualized dietary components, concluded that it could be effective for the induction and maintenance of remission in patients with mild to moderate CD [77]. The ESPEN guideline on Clinical Nutrition in Inflammatory Bowel Disease recommendations indicates, regarding adulthood, that the CDED should be considered, with or without partial enteral nutrition (PEN) [78,79]. Due to its composition, the CDED can be considered an anti-inflammatory diet [79].

One MA on dietary interventions for UC (adults and children) supported dietary interventions to maintain clinical remission, with a low or very low level of evidence [80].

A quality analysis of the existing guidelines on nutrition and diet for IBD revealed that they provide few recommendations and 50% of them support low-quality evidence [81].

Currently, the usefulness of diets with a low inflammatory potential in combination with biological therapy is being researched, published, and confirmed [82]. This practice can improve the low level of evidence achieved thus far in trials comparing diets alone.

Supplementation regimens should be controlled for diets in clinical trials to accurately compare interventions, which is not common practice. Adapting a diet to a patient’s needs is essential, considering unique characteristics when necessary (intolerances, malabsorption, intestinal strictures, and stenosis) [78].

Obtaining the same level of evidence in dietary trials as in pharmacological trials is challenging because of the study design and patient differences (microbiota, intolerances, allergies, BMI, opinions, level of adherence, comorbidities, and pharmacological treatment). These factors justify the need to individualize diets. The patient’s energy needs and the inflammatory index of food should also be considered.

### 4.2. Measurement of the Inflammatory Effect of Different Components of the Diet

Usually, the studies did not use an index that would allow the composition of the diets to be objectively compared.

The additive effect of different diet components should be investigated to improve study methodologies [44]. The composition of diets in the meta-analyses was reviewed, finding that most studies did not use the DII scoring algorithm. It would be interesting to compare the DII of the diets in the studies that are carried out, to evaluate the heterogeneity.

In this review, we found that diets do not have a standard name for similar compositions, and sometimes diets with different compositions are associated with the same global name. Mediterranean, Modified-Mediterranean, and similar Mediterranean diets were included in the same group in one meta-analysis [69]. It may be advisable to consider the different inflammatory potentials of these three diets. In this study, an anti-inflammatory diet was compared in a separate group. In addition, the network MA [69] simultaneously reviewed all diets in a single analysis by combining direct and indirect evidence. At present, the effectiveness of network MA for validating all possible indirect comparisons may be questionable in providing robust evidence because, in general, RCTs conducted on diets are not blinded, with low adherence and heterogeneity (differences in baseline patient characteristics, diet compositions, controls, countries with various dietary factors, and diet duration). However, we observed that Cochrane reviews, with a very good methodology, are very strict when comparing diets with the same name, obtaining very few patients for comparison; therefore, it is difficult to obtain conclusions.

### 4.3. DII May Be Correlated with Circulating Inflammatory Marker Levels

The MA by Kheirouri in 2019 [83] on the dietary inflammatory potential and risk of neurodegenerative diseases concluded that the DII may be correlated with circulating inflammatory marker levels. This confirms that the DII is an appropriate tool for measuring the inflammatory potential of a diet. The role of diets with an inflammatory potential in the pathophysiology of neurodegenerative diseases has been validated [83,84].

The 2024 findings of Rad SR-MA on DII and MS/demyelinating autoimmune diseases suggest that a pro-inflammatory diet may increase the odds of developing these pathologies. An association between higher DII scores and the likelihood of developing MS has also been found [85].

The findings by Mirhosseini et al., SR (2023) suggest that the anti-inflammatory diet as measured by lower DII scores was associated with a better gut microbiome profile and variations in the composition and variety of the microbiome [86].

The DII of the diets is not usually known in diet studies. In addition, only three of the eight reviewed meta-analyses comparing potentially anti-inflammatory diets studied inflammatory biomarkers. However, the reduction in inflammatory biomarker levels was unclear in those meta-analyses.

### 4.4. Restriction of Common Components in Many Diets

A systematic review by Lerner et al., with very low levels of evidence, concluded that a gluten-free diet (GFD) could be useful for treating non-gluten-dependent ADs, reducing symptoms in many patients, and suppressing harmful intraluminal intestinal events [87]. Patients with rheumatic diseases present gluten sensitivity more frequently than the general population. Some RCTs included in the meta-analyses investigated modified Mediterranean diets with gluten reduction [62].

An association between psoriasis, celiac disease, and celiac disease markers has previously been identified. According to preliminary studies, a GFD may benefit some patients with psoriasis [88,89].

A low-FODMAP diet can be considered an anti-inflammatory diet because it includes the withdrawal of potentially inflammatory foods, such as fructose (sugar), sorbitol or mannitol (derived from sugar), lactose (dairy), fructans (wheat, rye, and oats), and galactans (chickpeas, rye, and oats). However, after symptom improvement with a low-FODMAP diet, the diet should be modified because it eliminates some prebiotics that can damage the IM [90].

Likewise, wheat contains fructans (FODMAPs), therefore the GFD may be lower in FODMAPs; moreover, the continuous use of a GFD can cause deficiencies in minerals, vitamins, and Fiber. Foods manufactured without gluten are usually rich in salt, fat, and sugar, which can contribute to inflammation by modifying IM. Currently, designing an individualized diet has been proposed as a strategy to modulate the composition of IM [91]. Therefore, nutrition specialists should supervise diets.

The CDED, specifically designed for CD patients, is a whole food diet combined with PEN and excluding processed foods, emulsifiers, animal fat, red and processed meat, carrageenan, gluten, and dairy products among others [77,79]. Owing to its composition, we could consider the CDED to be an anti-inflammatory diet, confirming our findings. Likewise, modified Mediterranean diets tend to reduce the intake of inflammatory foods [68].

This review demonstrates the need for standardization in dietary research, to obtain higher-quality studies and consequently higher levels of evidence. It would be useful to compare anti-inflammatory diets using the DII, E-DII or another consensus index. This practice could be applied in future clinical trials and meta-analyses, and perhaps also retrospectively to validate results.

Given the characteristics of the current state of dietary research at IMID, we do not consider it advisable to use network meta-analysis for the overall analysis of the data. In order to use this method, a prior standardization of diets would be necessary as a first step.

Clinicians should be aware of the role of diets with anti-inflammatory properties as a complement to pharmacological treatments in IMIDs, taking into account that at present, although the evidence on diets with specific components with anti-inflammatory capacity is of low level, under current conditions the same level of evidence of drug trials is not achievable, although their potential is unquestionable.

## 5. Conclusions

Reduced pain, fatigue, and quality of life were observed in the patients with IMIDs and specific dietary interventions, mostly with anti-inflammatory properties due to their components, with critically low or low levels of evidence.

This review summarizes the global dietary evidence to improve IMID symptoms, discusses the influences of dietary mechanisms, clarifies the weaknesses of clinical trials and dietary meta-analyses, with critically low or low levels of evidence, and shows the need to use indices such as DII, which allow diets to be classified according to their content of pro-inflammatory or anti-inflammatory foods, to better compare diet groups in clinical trials. The difficulty of obtaining high-level evidence from dietary studies is apparent and may delay the application of the results. Clinicians should be aware of the role of diets with anti-inflammatory properties as a complement to pharmacological treatments in IMIDs.

## Figures and Tables

**Figure 1 nutrients-17-00493-f001:**
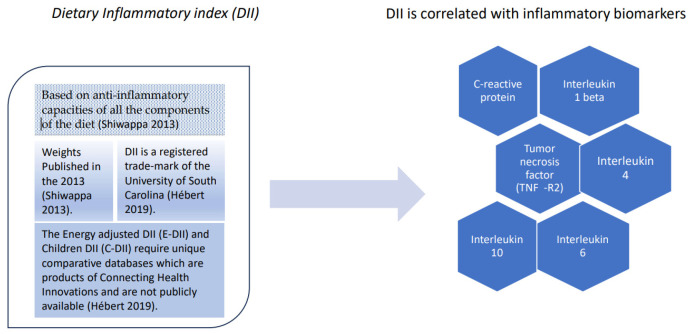
Dietary inflammatory index (DII) [38,39].

**Figure 2 nutrients-17-00493-f002:**
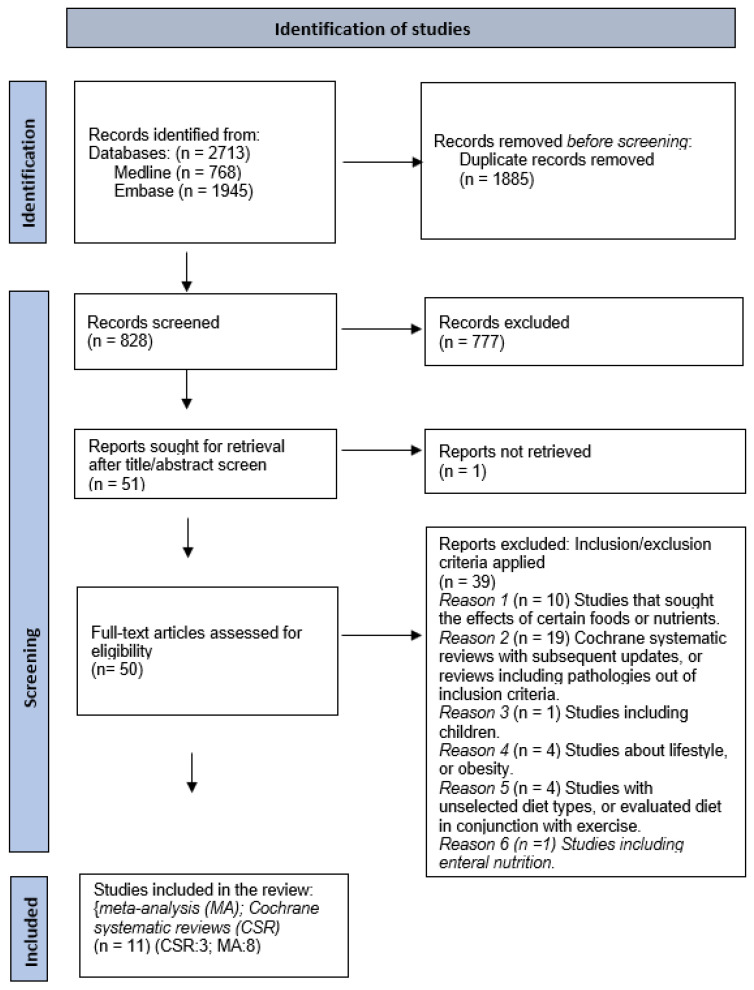
PRISMA. Flow diagram.

**Table 1 nutrients-17-00493-t001:** Information about diet components and positive or negative results in some immune-mediated inflammatory diseases.

Component	Disease	Results
Dietary fiber	-	The high-fiber food improves IM [8].
	RA	Can help reduce joint pain [9].
	CD, RA	Improve remission rates [9,10].
Omega-3 polyunsaturated fatty acids	-	The microbiota modulates the gut immune system through omega-3 polyunsaturated fatty acids, maintaining gut health [11].
	RA	A low omega-3/omega-6 fatty acid ratio promotes inflammation and increases the risk of RA [9].
Vitamin D	RA, MS	Deficiency reported as a risk factor [5].
Sourdough fermentation	-	Reduces pro-inflammatory activity [12].
Shelled fruits	-	A higher intake of shelled fruits correlated with lower levels of IL-6, a study in HP (LGCI) [13].
High carbohydrate intake	RA	Reducing dietary carbohydrates can help improve the balance between IM and immune function [9].
	IBD	The onset of IBD is associated with high carbohydrate intake, including sugar [14].
Sweets		IL-8 levels were increased with the frequent intake of sweets -A study in HP- (LGCI) [13].
Red meat		Red and processed meats, correlated with alterations in the IB [15]
		Red meat was associated with an inflammatory pattern, characterized by an increase in IL-6 and IL-8 levels -A study in HP- (LGCI) [13].
	RA	Excessive consumption of red meat can lead to changes in the microbiota, and increase the risk of RA [16].
Dairy	IBD Psoriasis	Excessive consumption may increase the risk of acute flares [15].
Wheat and other cereals	ADs	Contribute to chronic inflammation and ADs by increasing intestinal permeability [17].
Alcohol and food additives (sweeteners, emulsifiers, advanced glycation end products)		IB can be negatively affected [18].
High-sodium diet		Can lead to dysbiosis [9].
High-fat diet	UC	IB dysfunction -dysregulated microbiota and metabolites- [19].

IM: Intestinal microbiota. IB: Intestinal Barrier. RA: Rheumatoid arthritis. CD: Chron Disease. IBD: Inflammatory bowel disease. ADs: Autoimmune diseases. UC: Ulcerative Colitis. LGCI: Low-grade chronic inflammation. HP: Healthy people.

## Data Availability

The original contributions presented in the study are included in the article/Appendix A.

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
