# Peer review of "Specific Composition Diets and Improvement of Symptoms of Immune-Mediated Inflammatory Diseases in Adulthood—Could the Comparison Between Diets Be Improved?"

_nutrients, 2025, doi:10.3390/nu17030493_

Round 1
Reviewer 1 Report (New Reviewer)
Comments and Suggestions for Authors
This manuscript attempts to address the important relationship between dietary interventions and immune-mediated inflammatory diseases (IMIDs). While the topic is both timely and relevant, the current presentation significantly undermines its potential contribution to the field.
The manuscript's most immediate concern lies in its structural organization and presentation. The flow of information feels disjointed, often reading more like a collection of loosely connected segments rather than a cohesive scientific review. This is particularly evident in the Results section, where findings are presented in a somewhat haphazard manner without a clear narrative thread connecting the various studies discussed.
The PRISMA methodology, while cited, has been implemented rather superficially. The flow diagram, which should serve as a clear roadmap of the study selection process, appears hastily constructed and lacks professional polish. This is not merely a cosmetic issue - it reflects a broader pattern of methodological imprecision that permeates the manuscript.
The authors' treatment of statistical analyses is particularly concerning. While meta-analyses are presented, they lack the depth and rigor one would expect in a comprehensive systematic review. The absence of proper forest plots and detailed heterogeneity analyses weakens the validity of the conclusions drawn. Moreover, the tables, which should serve as clear repositories of data, are poorly formatted and often difficult to interpret.
The narrative synthesis of evidence also requires substantial development. The current presentation often fails to draw meaningful connections between studies or to critically evaluate the quality of evidence being presented. This results in a somewhat superficial treatment of what is actually a complex and nuanced topic.
However, it's important to note that these shortcomings do not entirely negate the manuscript's potential value. The authors have clearly invested considerable effort in gathering relevant literature, and the basic framework for a meaningful contribution exists. With substantial revision, this could develop into a valuable addition to the literature.
To strengthen the manuscript, several key improvements would be beneficial:
First, the overall structure needs tightening, with a more logical flow of information and clearer connections between sections. The methodology section particularly needs expansion, with detailed documentation of search strategies and selection criteria.
Second, the statistical analyses require substantial enhancement. Proper forest plots should be included, and heterogeneity should be more thoroughly addressed. The tables need professional reformatting to enhance clarity and accessibility.
Third, the narrative synthesis needs deeper development, with more critical evaluation of the evidence and clearer articulation of the relationships between different studies and findings.
Finally, the discussion section would benefit from a more nuanced treatment of the limitations and implications of the findings. The current treatment feels somewhat superficial and could be expanded to provide more meaningful insights for both researchers and clinicians.
Looking forward, this manuscript has the potential to make a valuable contribution to our understanding of dietary interventions in IMIDs. The basic elements are present, but significant revision is needed to realize this potential. The authors should be encouraged to undertake these revisions, as the topic is both important and timely.
The manuscript, in its current form, presents fundamental flaws in methodology, statistical analysis, and overall presentation that render it unsuitable for publication in Nutrients or any comparable peer-reviewed journal. While the underlying research question has merit, the manuscript requires substantial revision to meet basic academic publishing standards. The authors are encouraged to undertake a complete restructuring of their work, with particular attention to methodological rigor, statistical analysis, and professional presentation, before resubmission can be considered.
Comments on the Quality of English Language
The manuscript exhibits concerning issues with English language usage that significantly impact its readability and academic
- The text fluctuates between passages of adequate academic English and sections that appear poorly translated or drafted
- Some sentences show clear evidence of non-native construction, possibly indicating multiple authors with varying English proficiency
- Several paragraphs display awkward phrasing that obscures their scientific content
- Inconsistent verb tense usage throughout the manuscript
- Improper article usage (missing or incorrect use of "the" and "a/an")
- Run-on sentences that compromise clarity
- Comma splices and inappropriate punctuation
- Non-idiomatic expressions that suggest direct translation from another language
- Scientific terminology is not consistently applied
- Complex concepts are often expressed in overly simplistic or unclear language
- Lack of proper transitional phrases between sections
- Academic register is not maintained throughout
The manuscript requires comprehensive English language editing by a native speaker with expertise in academic/scientific writing before it can be considered for publication. The current language quality falls below the standard expected for a peer-reviewed international journal.
Author Response
Reviewer1
We greatly appreciate your contribution, and respond to your proposals.
“This manuscript attempts to address the important relationship between dietary interventions and immune-mediated inflammatory diseases (IMIDs). While the topic is both timely and relevant, the current presentation significantly undermines its potential contribution to the field”.
- We understand that it is difficult to achieve the quality of a Cochrane review in our presentation, considering that the objective is ambitious and sets of studies already analyzed in previous reviews are reviewed.
The manuscript's most immediate concern lies in its structural organization and presentation. The flow of information feels disjointed, often reading more like a collection of loosely connected segments rather than a cohesive scientific review. This is particularly evident in the Results section, where findings are presented in a somewhat haphazard manner without a clear narrative thread connecting the various studies discussed.
- The studies analyzed cannot be related because diets in general do not have clear characteristics to be compared with each other or to be analyzed jointly. In fact, the objective of our work is to provide evidence of this problem so that in the future this does not occur when diets are compared in different pathologies.
The PRISMA methodology, while cited, has been implemented rather superficially. The flow diagram, which should serve as a clear roadmap of the study selection process, appears hastily constructed and lacks professional polish. This is not merely a cosmetic issue - it reflects a broader pattern of methodological imprecision that permeates the manuscript.
- We have tried to be practical, and perhaps that is why the searches have not been completely mechanized, but rather expressly designed to rationally select the articles. A peer review was carried out and a third party resolved any disagreements.
The authors' treatment of statistical analyses is particularly concerning. While meta-analyses are presented, they lack the depth and rigor one would expect in a comprehensive systematic review. The absence of proper forest plots and detailed heterogeneity analyses weakens the validity of the conclusions drawn. Moreover, the tables, which should serve as clear repositories of data, are poorly formatted and often difficult to interpret.
- To jointly analyze data using statistical procedures, it is necessary to have quality baseline data that can be treated jointly. As we try to show throughout the study, this is not normal due to the difference between the diets being compared, the lack of data that allows comparing similar controls…
The narrative synthesis of evidence also requires substantial development. The current presentation often fails to draw meaningful connections between studies or to critically evaluate the quality of the evidence being presented. This results in a somewhat superficial treatment of what is a complex and nuanced topic.
- We agree that the result gives a somewhat superficial treatment of what is a complex and nuanced topic. But we cannot stop moving forward to find a solution to this. Our study does not have grand ambitions, only to provide a clear view of what can be improved in diet research in immune-mediated inflammatory diseases so that we can move forward.
However, it's important to note that these shortcomings do not entirely negate the manuscript's potential value. The authors have clearly invested considerable effort in gathering relevant literature, and the basic framework for a meaningful contribution exists. With substantial revision, this could develop into a valuable addition to the literature.
- We appreciate your positivity
“To strengthen the manuscript, several key improvements would be beneficial:
First, the overall structure needs tightening, with a more logical flow of information and clearer connections between sections. The methodology section particularly needs expansion, with detailed documentation of search strategies and selection criteria”.
- The search strategies are shown in Supplementary Table 1, and the selection criteria in the methodology
Second, the statistical analyses require substantial enhancement. Proper forest plots should be included, and heterogeneity should be more thoroughly addressed. The tables need professional reformatting to enhance clarity and accessibility.
- We consider that, in general, the characteristics of the patients and the diets of the intervention and control groups are not sufficiently defined to be able to compare them using forest plots. Therefore, we think that the diets of the intervention groups should have a similar and defined dietary inflammatory index, which would serve as a basis for making these comparisons, as well as the diets of the control groups among themselves.
Third, the narrative synthesis needs deeper development, with more critical evaluation of the evidence and clearer articulation of the relationships between different studies and findings.
- As we mentioned previously, the articulation that is requested of us would be of poor quality due to not having a basic dietary inflammatory index.
Finally, the discussion section would benefit from a more nuanced treatment of the limitations and implications of the findings. The current treatment feels somewhat superficial and could be expanded to provide more meaningful insights for both researchers and clinicians.
- We have made an effort in the discussion and we consider that we cannot provide more meaningful insights
Looking forward, this manuscript has the potential to make a valuable contribution to our understanding of dietary interventions in IMIDs. The basic elements are present, but significant revision is needed to realize this potential. The authors should be encouraged to undertake these revisions, as the topic is both important and timely. The manuscript, in its current form, presents fundamental flaws in methodology, statistical analysis, and overall presentation that render it unsuitable for publication in Nutrients or any comparable peer-reviewed journal. While the underlying research question has merit, the manuscript requires substantial revision to meet basic academic publishing standards. The authors are encouraged to undertake a complete restructuring of their work, with particular attention to methodological rigor, statistical analysis, and professional presentation, before resubmission can be considered.
- We would love to work with some initial studies that would allow us to do quality statistical analyses, that was our initial goal, but we consider that the results of analyses based on low-quality studies are never applied in clinical practice. An example: in terms of the level of evidence, comparing two studies of the same drug (for example, adalimumab), with the same dose in patients with the same pathology and the same characteristics, cannot be the same as comparing two diets that are called “Mediterranean” and that do not have the same composition, patient compliance is not known, intolerances and specific characteristics of the patients are not known…
Comments on the Quality of English Language
The manuscript exhibits concerning issues with English language usage that significantly impact its readability and academic
The manuscript requires comprehensive English language editing by a native speaker with expertise in academic/scientific writing before it can be considered for publication. The current language quality falls below the standard expected for a peer-reviewed international journal.
We have submitted the manuscript to Elsevier language reviewers twice, first in American English, and then in British English.
Reviewer 2 Report (New Reviewer)
Comments and Suggestions for Authors
the article is very important nowadays, because the microbiota is essential for developing and relapsing the inflammatory diseases.
The introduction is clearlly described, a small recomendation is to add a figure with all the diet principles (adding the food groups for each) and highlighting the differences in the conclusion..
The material and methods are well designed, the articles are very recent.. and useful.
the Results are results are described, not just presented in a table, and in the discussion chapter the analysis is extensive
the conclusion is clear: the physicians have to discuss with their patients even about diet, emotional stress not only about medications and their possible adverse events.. a comprehensive medicine....
Author Response
We greatly appreciate your contributions as they have been very useful to us.
Nowadays, the article is very important, because the microbiota is essential for developing and relapsing Inflammatory diseases. The introduction is clearly described, a small recommendation is to add a figure with all the diet principles (adding the food groups for each) and to highlight the differences in the conclusion
The material and methods are well designed, The articles are very recent .. and useful.
the Results are results are described, not just presented in a table, and in the discussion chapter the analysis is extensive
the conclusion is clear: the physicians have to discuss with their patients even about diet, Emotional stress not only about medications and their possible adverse events.. a comprehensive medicine....
We have added a new summary table on specific dietary components, the current Table 1. Information about diet components and positive or negative results in some immune-mediated inflammatory diseases.
Additionally, we have concluded by modifying the paragraph:
“A reduction in pain, fatigue and quality of life was observed in patients with IMIDs and specific dietary interventions, generally with anti-inflammatory properties due to their components, with critically low or low levels of evidence.”
We have also added this conclusion to the abstract.
Reviewer 3 Report (New Reviewer)
Comments and Suggestions for Authors
The authors aimed to review the quality of evidence on the effects of specific diet composition on symptoms of immune-mediated inflammatory diseases (IMIDs), including rheumatoid arthritis (RA), spondyloarthritis, multiple sclerosis (MS), inflammatory bowel disease (IBD) [remission maintenance of Crohn´s disease and ulcerative colitis], psoriasis and psoriatic arthritis in adult patients. The structure of the manuscript is well organized. However, more should be said about inflammatory cytokines and specific diets in the introduction. This part seems to be missing. To make this review more appealing to readers, you should include a schematic illustration.
Author Response
We greatly appreciate your contributions as they have been very useful to us.
The authors aimed to review the quality of evidence on the effects of specific diet composition on symptoms of immune-mediated inflammatory diseases (IMIDs), including rheumatoid arthritis (RA), spondyloarthritis, multiple sclerosis (MS), inflammatory bowel disease (IBD) [remission maintenance of Crohn´s disease and ulcerative colitis], psoriasis and psoriatic arthritis in adult patients. The structure of the manuscript is well organized. However, more should be said about inflammatory cytokines and specific diets in the introduction. This part seems to be missing. To make this review more appealing to readers, you should include a schematic illustration
We have included two paragraphs with information on cytokines in the introduction. New citations 34 and 35 from Nutrients on cytokines and diet have been included in the bibliography. We have also added a new summary table on specific dietary components, the current Table 1. Information about diet components and positive or negative results in some immune-mediated inflammatory diseases. Some information on cytokines is also included in this table.
Round 2
Reviewer 1 Report (New Reviewer)
Comments and Suggestions for Authors
After reviewing the exchange between the reviewer and authors, I find myself in a nuanced position. The manuscript tackles an important topic - how dietary interventions affect immune-mediated inflammatory diseases - but its execution needs substantial improvement.
The authors make some valid points in their response, particularly about the inherent challenges of comparing dietary studies with the same rigor as pharmaceutical trials. They correctly note that the heterogeneity of dietary interventions makes traditional meta-analysis techniques problematic. Their core argument about needing standardization through measures like the Dietary Inflammatory Index (DII) is sound and potentially valuable to the field.
However, their response falls short in several areas. They've been somewhat defensive about structural and methodological criticisms rather than embracing the opportunity to strengthen their paper. The disjointed presentation of results, which the reviewer highlighted, isn't adequately addressed in their response. Their answer about language revision is particularly thin - simply mentioning Elsevier language review without further detail isn't sufficient for an academic publication.
Looking forward, I believe this paper could make a meaningful contribution with substantial revision. The authors need to focus on better organizing their material, creating clearer narrative flow, and more explicitly acknowledging limitations while maintaining their important central argument about standardization needs in dietary research.
I would recommend they:
- Completely restructure the paper to create better flow
- Expand their methodology section, particularly regarding PRISMA implementation
- Provide clearer presentation of results, even if traditional meta-analysis isn't possible
- Strengthen their narrative synthesis
- Seek professional language editing
- Better integrate various components of the review
- More explicitly acknowledge limitations
- Provide clearer practical implications for researchers and clinicians
The authors' point about the difficulty of comparing dietary studies with pharmaceutical trial rigor is well-taken, but they need to make this case more effectively within a well-structured paper. They should emphasize how their review demonstrates the need for standardization in dietary research while presenting their findings in a more organized and accessible way.
While the reviewer's criticisms about statistical analysis and forest plots are valid from a traditional systematic review perspective, the authors make a reasonable case for why such approaches might not be appropriate given the current state of dietary research. However, they need to make this argument more explicitly and clearly in their paper, perhaps suggesting alternative ways to present and analyze their findings.
The language issues must be addressed more comprehensively. A thorough review by a native English-speaking academic editor would be essential before resubmission. The tables need professional reformatting, and the PRISMA flow diagram requires more careful attention to detail.
In conclusion, while the manuscript currently falls short of publication standards, it has the potential to make a valuable contribution to the field. The authors need to embrace a major revision while maintaining their important insights about standardization needs in dietary research for IMIDs. With careful attention to structure, clarity, and presentation, along with professional language editing, this could become a useful addition to the literature. The key is to address the reviewer's concerns constructively while better showcasing the unique value of their analysis.
Comments on the Quality of English Language
The English could be improved to more clearly express the research.
Author Response
We greatly appreciate your contributions as they have been very useful to us. We have made some changes to suit your requirements:
We regret not having followed your advice from the beginning, but we initially thought that it could not be put into practice. Initially we understood that the bibliographic search was poorly done and that we had to start again. We have subsequently calmly analyzed the prism diagram and corrected some errors in incorrectly transcribed counts.
Something similar happened with the suggested modifications to the methodology. We understood that it had to be modified, and after working for so long, it was too much for us to handle. Now we understand that it is about writing in a different way and we have tried to improve the wording. We have rewritten part of the methodology.
We greatly value the reviewer's knowledge and analysis. Structure and clarity are factors that have taken us a lot of work to improve. It has been difficult and we have reviewed them many times, but we have given it another go and made corrections. New supplementary tables are provided to make the search clearer and the manuscript easier to understand.
English and writing have been re-checked, and some errors have been corrected
We have tried to strengthen the narrative synthesis and connecting the results, for example in the introduction:
"We have summarized the results of the influence of specific dietary components on the reduction of IMID symptoms. However, the analysis of the practical application of these results using diets with specific composition is still evolving and, due to the great heterogeneity of the studies, it has been difficult in the past to draw concrete conclusions, with sufficient evidence to be applied."
We have emphasized how the review demonstrates the need for standardization in dietary research, as suggested by the reviewer, and provide clearer practical implications for researchers and clinicians, at the end of the discussion, and conclusions:
This review demonstrates the need for standardization in dietary research, to obtain higher quality studies and consequently higher levels of evidence. It would be useful to compare anti-inflammatory diets using the DII, E-DII or another consensus index. This practice could be applied in future clinical trials and meta- analyses, and perhaps also retrospectively to validate results.
Given the characteristics of the current state of dietary research at IMID, we do not consider it advisable to use network meta-analysis for the overall analysis of the data. In order to use this method, a prior standardization of diets would be necessary as a first step.
Clinicians should be aware of the role of diets with anti-inflammatory properties as a complement to pharmacological treatments in IMIDs, taking into account that at present, although the evidence on diets with specific components with anti- inflammatory capacity is of low level, under current conditions the same level of evidence of drug trials is not achievable, although their potential is unquestionable.
The limitations are explicitly summarized in the first section of the discussion (4.1).
Reviewer 3 Report (New Reviewer)
Comments and Suggestions for Authors
fine
Author Response
Thank you very much
This manuscript is a resubmission of an earlier submission. The following is a list of the peer review reports and author responses from that submission.
Round 1
Reviewer 1 Report
Comments and Suggestions for Authors
Based on the title, the question raised is NO, and this “rapid review” adds very little to help address diets and whether some do or do not affect IMIDs. There are no Tables to connect any diet with any IMID. Many of the statements throughout the manuscript would require the reader to go to the referenced article to get a relevant idea of any dietary influence on any IMID. Some of the deficiencies may be due to inadequate descriptions or poor wording. Figure 1 describing the records used and those excluded needs information and reasoning for exclusion. Overall, the paper needs some tables making some of the points about diet and IMID. The reader now has to go to the reference cited to get better idea of what is being presented. Examples follow:
Line 40-41: “In addition to the increase in frequency, ADs are often co-associated, and 25% of patients with these diseases develop other comorbid ADs.” Examples or more specifics would be helpful.
Line43: “more organs” meaning more than one organ with inflammation?
Line 45: likely should be and/or
Line 50: “cause” would require more evidence, can exacerbate might be better.
Line 55-56: description of microbiome not microbiota.
Line 57: immune disorders involve more than just T cells. This needs to be described more fully and accurately.
Line 60-61: Three reviews but only one reference (9) is given
Line 62: “RA …and other neurological disorders” RA is not usually listed as being neurological.
Line 78: “High glucose and fructose intake can reduce microbial diversity” this needs a reference more data to support the statement.
Line 88: ” microbiota modulates the gut immune system through omega-3” with ingestion of
Line 100: increased inflammation from the systemic entrance of gut microbes
Line 101: why is only vitamin D mentioned? Folate and B vitamins are just as important
Line 114: “The immune system contains the highest energy-consuming cells in the body” I believe neuroscientists would disagree.
Line 137-138: Why are only these six inflammatory biomarkers listed? IL-10 is often considered anti-inflammatory
Line 270, 271, 283 mention Table 1 and Table 2 but the manuscript has no Tables.
Line 386: “modulate the immune system” this needs more information
Line 392: “The limitation of this global review is that the meta-analyses included studies with 392 small samples and were predominantly of poor quality.” This comment describes the inadequacies of this manuscript.
Line 407: “works in favor” unclear sentence
Line 437: “and confirmed” with regard to what?
Line 449: “Usually, the studies did not analyze an index that would allow the composition of the diets to be objectively compared.” This is true for this report as well.
Conclusions: Like most of the report, there is a lack of clarity and inadequate descriptions to allow any useful update on dietary influences on IMID. There are no clear descriptions of any dietary component being mechanistically connected to an IMID.
Comments on the Quality of English Language
In general, English is OK but the way some sentences are worded it is unclear what point is being made
Author Response
We greatly appreciate your contributions as they have been very useful to us.
When we initially sent the article to "Nutrients", we created a manual version, and later we created a different one on the MDPI platform. We have verified that this MDPI version was possibly the one that the reviewers used. We have verified that this initial MDPI version does not contain tables 1 and 2.
In the new version of the article, in addition to Figure 1, and Tables 1 and 2, two new Supplementary Tables (2 and 3) have been added, with the characteristics of the studies analyzed in Systematic Reviews, and Meta-Analyses, to increase the quality and facilitate your understanding.
We have made some changes to suit your requirements:
- Based on the title, the question raised is NO, and this “rapid review” adds very little to help address diets and whether some do or do not affect IMIDs
We have made an effort to improve the writing and we have simplified the title, because in addition to summarizing real evidence and to show the weaknesses common to most studies, what we want to show is some way to improve them. We already know about the difficulty of achieving quality in diet studies and, therefore, we are aware of the need to show how to solve it.
- There are no Tables to connect any diet with any IMID. Many of the statements throughout the manuscript would require the reader to go to the referenced article to get a relevant idea of any dietary influence on any IMID. Some of the deficiencies may be due to inadequate descriptions or poor wording. Overall, the paper needs some tables making some of the points about diet and IMID. The reader now has to go to the reference cited to get a better idea of what is being presented.
In the new version of the article, in addition to the modified Figure 1 and Tables 1 and 2 and Supplementary Table 1, two new supplementary tables (2 and 3) are provided with the data from the studies analyzed in the systematic reviews and meta-analyses, to increase the quality and facilitate their understanding.
- Figure 1, describing the records used and those excluded needs information and reasoning for exclusion.
In Figure 1 we have included the reasons for exclusion.
- Conclusions: Like most of the report, there is a lack of clarity and inadequate descriptions to allow any useful update on dietary influences on IMID. There are no clear descriptions of any dietary component being mechanistically connected to an IMID.
The introduction to the study was not intended to provide an exhaustive update on the mechanistic connection of dietary components with (IDID). The only intention in writing the introduction was to provide a reliable summary of the information, which would allow us to see the need to objectively classify diets based on the pro- or anti-inflammatory capacity of the set of their components, and that the studies should be able to be compared knowing the quantitative difference in inflammatory power between the diets of the intervention and control groups, with some index that would allow diets not to be compared only by their name, but by analyzing the effect of the union of their different dietary components. Also, to avoid any confusion, we have changed the title of the article.
Minor concerns:
We have resolved the minor concerns as well.
Reviewer 2 Report
Comments and Suggestions for Authors
This study aimed to review the evidence on specific diet composition effects on some immune-mediated inflammatory diseases. But the quality of the manuscript is poor.
Major concerns:
1. The writing is logically disorganized and cut into small paragraphs, making the main idea unclear and difficult to understand.
2. The limitation of this review is that the meta-analyses included studies with small samples and were predominantly of poor quality, which leading to a lack of clear results from the study.
3. There are several references to TABLE 1 in the manuscript, but TABLE 1 is not found in the manuscript.
Minor concerns:
1. Line 162, An additional bracket has been added to the end of the paragraph.
2.Line 253, Missing a comma between “biomarkers” and “One”.
3.Line 382, “COCHRANE SR” should be revised to “Cochrane SR”.
Comments on the Quality of English Language
The quality of English Language should be further improved.
Author Response
We greatly appreciate your contributions as they have been very useful to us.
When we initially sent the article to "Nutrients", we created a manual version, and later we created a different one on the MDPI platform. We have verified that this MDPI version was possibly the one that the reviewers used. We have verified that this initial MDPI version does not contain tables 1 and 2.
We have made some changes to suit your requirements:
- “The writing is logically disorganized and cut into small paragraphs, making the main idea unclear and difficult to understand”.
We have tried to improve the writing of the manuscript and have asked the English technicians at Elsevier to review the manuscript (we requested the British version because it was previously submitted to "Nutrients" in the American language and was not considered correct).
- The limitation of this review is that the meta-analyses included studies with small samples and were predominantly of poor quality, which leading to a lack of clear results from the study.
The clinical trials and meta-analyses that have been published on diets in immune-mediated inflammatory diseases cannot reach a high level of quality, and in fact, this study attempts to find areas for improvement. We included this limitation in the writing of the article.
- There are several references to TABLE 1 in the manuscript, but TABLE 1 is not found in the manuscript.
Table 1 is definitive for understanding the manuscript, it contains the characteristics of the studies and the results of the values ​​of the meta-analyses.
In the new version of the article, in addition to Figure 1, tables 1 and 2, which due to our mistake did not appear in MDPI, and supplementary table 1, two new supplementary tables (2 and 3) are provided with the data of the studies evaluated in the Systematic reviews, and Meta-analyses, to increase the quality and facilitate their understanding.
Minor concerns:
We have resolved the minor concerns as well.